# Identification of the Core Promoter and Variants Regulating Chicken *CCKAR* Expression

**DOI:** 10.3390/genes13061083

**Published:** 2022-06-18

**Authors:** Zhepeng Wang, Angus M. A. Reid, Peter W. Wilson, Ian C. Dunn

**Affiliations:** 1College of Animal Science and Technology, Northwest A&F University, Yangling 712100, China; 2Royal (Dick) School of Veterinary Studies, Roslin Institute, University of Edinburgh, Midlothian EH25 9RG, UK; angus.reid@ed.ac.uk (A.M.A.R.); peter.wilson@roslin.ed.ac.uk (P.W.W.); ian.dunn@roslin.ed.ac.uk (I.C.D.)

**Keywords:** chicken, *CCKAR*, satiety, growth, core promoter, expression activity

## Abstract

Decreased expression of chicken cholecystokinin A receptor (*CCKAR*) attenuates satiety, which contributes to increased food intake and growth for modern broilers. The study aims to define the core promoter of *CCKAR*, and to identify variants associated with expression activity. A 21 kb region around the *CCKAR* was re-sequenced to detect sequence variants. A series of 5′-deleted promoter plasmids were constructed to define the core promoter of *CCKAR*. The effects of sequence variants located in promoter (PSNP) and conserved (CSNP) regions on promoter activity were analyzed by comparing luciferase activity between haplotypes. A total of 182 variants were found in the 21 kb region. There were no large structural variants around *CCKAR*. pNL−328/+183, the one with the shortest insertion, showed the highest activity among the six promoter constructs, implying that the key cis elements regulating *CCKAR* expression are mainly distributed 328 bp upstream. We detected significant activity differences between high- and low-growth associated haplotypes in four of the six promoter constructs. The high-growth haplotypes of constructs pNL−1646/+183, pNL−799/+183 and pNL−528/+183 showed lower activities than the low-growth haplotypes, which is consistent with decreased expression of *CCKAR* in high-growth chickens. Lower expression of the high-growth allele was also detected for the CSNP5-containing construct. The data suggest that the core promoter of *CCKAR* is located the 328 bp region upstream from the transcription start site. Lower expression activities shown by the high-growth haplotypes in the reporter assay suggest that CSNP5 and variants located between 328 bp and 1646 bp upstream form a promising molecular basis for decreased expression of *CCKAR* and increased growth in chickens.

## 1. Introduction

Regulation of appetite is a homeostatic process featuring reciprocal shifts between hunger and satiety sensations in response to energy state, which normally results in an appropriate growth rate and adult bodyweight [1]. The arcuate nucleus of the hypothalamus is the control center for hunger and satiety in mammals [2]. Here, orexigenic neuropeptide Y/agouti-related protein neurons and anorectic pro-opiomelanocortin/cocaine- and amphetamine-regulated transcript neurons integrate peripheral signals from vagal afferents and circulating factors to regulate food intake and energy expenditure [3]. A series of neurohormonal signals that are produced by the gut and adipose tissue are involved in appetite control. Ghrelin is the only known peptide hormone that stimulates hunger [4]. In contrast, there are many hormones that are associated with satiety. Of these, leptin and cholecystokinin (CCK) are two well-established satiety signals. Leptin is produced by adipose tissue and acts directly at the arcuate of the hypothalamus to regulate long-term energy homeostasis [3]. The gut peptide CCK not only acts as a short-term satiety signal, but is a key signaling molecule responsible for long-term regulation of feeding and energy balance by interacting with leptin [5]. In birds, CCK signaling may play an enhanced role in appetite control, as limited expression of leptin in adipose tissue and autocrine/paracrine mode suggest that leptin may not serve as an adiposkine involved in nutritional feedback [6,7,8].

Cholecystokinin initiates satiety by binding to receptors on the vagus nerve and the hypothalamus [3]. More localized effects of CCK on digestive activities occur through binding to receptors throughout the gastrointestinal tract in mammals [9]. Two G-protein coupled receptors—CCKAR and CCKBR—are responsible for transduction of CCK signaling in the mammals [9,10]. CCKBR binds CCK and gastrin with almost equal affinities and stimulates gastric acid secretion [9]. CCKAR exhibits a 500-fold higher affinity for CCK than for gastrin in pancreatic acinar cells and is the primary receptor mediating satiety signals in the mammals [4,9]. Loss of CCKAR due to spontaneous mutations decreased satiety and increased food intake and obesity in humans and rats [11,12]. Administration of a CCKAR antagonist similarly produced an appetite-promoting effect in broilers, pigs and rats [13,14,15].

Our previous studies found that high-growth chickens expressed less *CCKAR* transcripts than low-growth chickens in an Advanced Intercross Line (AIL) [16]. This finding points to one molecular basis underlying increased appetite and growth rate in modern chickens. An allele-specific expression assay showed that the abundance of *CCKAR* transcripts arising from the low-growth-associated allele was 3.5-fold higher than that of the high-growth associated allele, implying that the differential expression of *CCKAR* is caused by *cis*-acting mutations instead of a *trans* action [16]. The A allele of G.420C > A, a SNP in the 5′UTR of the *CCKAR* gene, was associated with higher bodyweight and average daily gain of chickens, and was postulated to affect *CCKAR* expression through disrupting a YY1 binding site in the *CCKAR* promoter [17]. Other variants in exons and the downstream region of *CCKAR* were also reported to be associated with faster growth and higher feed efficiency of chickens, implying that there may be other *cis*-regulatory variants in this gene or adjacent regions affecting *CCKAR* expression and growth [16,18,19].

The aims of this study were to define the key region regulating *CCKAR* expression using serial 5′-deleted constructs, and to identify variants associated with expression activity by prediction analysis in silico of sequence variants and a reporter gene assay.

## 2. Materials and Methods

### 2.1. Re-Sequencing the CCKAR Locus

Four chickens were selected from generation 16 of the AIL, in which the *CCKAR* locus was identified as the largest QTL accounting for a 19% difference in bodyweight [16]. Variants at the *CCKAR* locus were found to be fixed into two haplotypes in the AIL. One associated with high growth was defined as the high-growth (HG) haplotype, and the other is named the low-growth (LG) haplotype [16]. Two of four birds carried the HG haplotype and the other two with the LGH haplotype. A 21 kb region (Galgal 6.0, chr4:73195458–73216805) flanking the *CCKAR* was re-sequenced. Genomic DNA was extracted from blood using DNAzol reagent (Thermo Fisher Scientific, Waltham, MA, USA) according to the manufacturer’s instructions. Thirty-five fragments covering the 21 kb region were amplified by PCR. PCR preparations comprised 2 μL of 10 × FastStart buffer, 2 μL of 10 × dNTP mix, 0.5 μL of forward primer (20 μM), 0.5 μL of reverse primer (20 μM), 0.1 μL of FastStart Taq polymerase (Roche, Basel, Switzerland), 1 μL of template gDNA and 13.9 μL of ddH_2_O. The PCR reaction was run in a *BIO-RAD* T100 thermal cycler (Bio-Rad, Hercules, CA, USA) under reaction conditions: 95 °C for 4 min, 40 cycles of (95 °C for 30 s, 58 °C for 30 s, 72 °C for 30 s), and 72 °C for 7 min. Primers were designed using Primer3. Primers and amplicons are detailed in Appendix A. After the specificity of PCR was verified by agarose gel electrophoresis, PCR products were bi-directionally sequenced using the Sanger sequencing method. Contiguous sequences were assembled using SeqBuilder (DNASTAR, Madison, WI, USA). The resulting sequences were deposited in the GenBank database (https://www.ncbi.nlm.nih.gov/Genbank/ (26 August 2020): accession number MT522011 for the HG haplotype and MT522012 for the LG haplotype. Genetic variants were called by multiple alignment using DNAMAN 6.0 (Lynnon BioSoft, San Ramon, CA, USA).

### 2.2. Identification of Conserved Element Variants

Sequence conservation of the 21 kb re-sequenced region was analyzed by aligning homologous regions of 76 vertebrate species consisting of 52 birds, 9 reptiles and 15 other species using the MULTIZ program in the UCSC database [20]. Conserved element variants were identified by the PhastCons program in the UCSC database [21]. *CCKAR* variants were displayed in the UCSC genome browser using the “Add Custom Tracks” tool in order to allow visual identification of conserved element variants (CSNP).

### 2.3. Construction of Reporter Gene Plasmids

To identify key *cis*-elements controlling *CCKAR* expression, five serial 5′-deletion and one intragenic fragments from the 5′ upstream region of *CCKAR* were fused into the upstream region of a nanoluciferase reporter gene (Figure 1). In addition, regulatory effect analysis of five SNPs (CSNP1–CSNP5) located in conserved elements was also included in this reporter assay. Serial 5′-deleted and CSNP-contained regions were amplified using the HG and LG templates. Primer sequences are listed in the Table 1. PCR products were purified using a QIAquick^®^ PCR Purification Kit (QIAGEN, Venlo, The Netherlands). Each purified PCR product and pNL1.1 stock plasmid were digested with *KpnI* and *XhoI* restriction endonucleases (New England BioLabs Inc., Ipswich, MA, USA). Digested products were purified using a QIAEX II Gel Extraction Kit (QIAGEN, Venlo, The Netherlands) according to the Quick-Start Protocol. Target fragments were fused into upstream regions of the nanoluciferase reporter gene of the pNL1.1 plasmid (Promega, Madison, WI, USA) using T4 ligase (Promega, Madison, WI, USA). These constructs were transformed into Subcloning Efficiency™ DH5α™ Competent Cells (Invitrogen, Waltham, MA, USA) by heat shock according to the manufacturer’s instructions. Transformants were incubated at 37 °C overnight on a selective LB agar plate containing 100 μg/mL ampicillin. Plasmid DNA was extracted from bacterial cultures made from single colonies using the QIAGEN Plasmid Midi kit (QIAGEN, Venlo, The Netherlands) according to the product handbook. Concentrations of plasmids were measured using a NanoDrop™ 2000 spectrophotometer (Thermo Fisher Scientific, Waltham, MA, USA). All plasmid constructs were verified by Sanger sequencing prior to use.

Five-prime deletion and one intragenic fragment fused into the upstream region of the nanoluciferase gene to define the core promoter for transcription and to analyze the regulatory effects of variants. A total of 19 SNPs were found in the −1646/+734 region and grouped into the high- and low-growth haplotypes (HG/LG) in the Advanced Intercross Line. Information on these SNPs is summarized in Appendix A.

### 2.4. Cell Culture, Transfection and Dual-Luciferase Reporter Assay

Chicken DF-1 cells were used for transient transfection of reporter plasmids. This cell line was kindly gifted by D. B. Zhao at the Roslin Institute, UK. DF-1 cells were seeded in 12-well culture plates at 3 × 10^4^ cells/well and cultured in high-glucose DMEM (Thermo Fisher Scientific, Waltham, MA, USA) supplemented with 1% (*v*/*v*) chicken serum (Sigma-Aldrich, St. Louis, MO, USA), 10% (*v*/*v*) fetal bovine serum (Sigma-Aldrich, St. Louis, MO, USA) and 1% (*v*/*v*) L-glutamine–penicillin–streptomycin (Sigma-Aldrich, St. Louis, MO, USA). At 70–80% confluence, target and pGL3-control (Promega, Madison, WI, USA) plasmids were co-transfected into DF-1 using Lipofectamine LTX reagent (Invitrogen, Waltham, MA, USA). One hundred microliters of transfection mixture was prepared for each well by diluting pNL1.1 constructs (1 μg) and pGL3-control plasmids (100 ng), 3 μL of Lipofectamine LTX and 2 μL of PLUS^TM^ reagent into Opti-MEM (Thermo Fisher Scientific, Waltham, MA, USA). After gentle mixing and incubation for 5 min at room temperature, 100 μL of transfection mixture was added per well into 1 mL of culture medium.

Dual-Luciferase reporter assay was performed 24 h after transfection using the Nano-Glo^®^ Dual-Luciferase Reporter Assay System (Promega, Madison, WI, USA) according to the product instructions. Luminescence intensity was detected using a LB-96V Microplate Luminometer (Berthold Technologies GmbH & Co. KG, Bad Wildbad, Germany). Empty pNL1.1 plasmid served as the negative control. Raw luminescence intensity values were divided by the input molar mass of each plasmid to normalized differences in plasmid size. The relative luciferase activity of each pNL1.1 construct was reported by dividing activity of each pNL1.1 construct by that of co-transfected pGL3-control plasmid. All reporter assays were repeated 6 times.

### 2.5. Predicting Disrupting Effects of Variants on TF Binding Motifs

The disrupting effects of variants on transcription factor (TF) binding motifs were predicted using the “Differential TF Binding” tool in CIS-BP database [22] under default conditions, with *Gallus gallus* selected as the species of interest for TF prediction. Effects of promoter variants on FOXO1 binding motifs were further predicted with 8 FOXO1-related entries deposited in the JASPAR database [23].

### 2.6. Statistical Analysis

Activities of promoter constructs were compared using ANOVA. The model for ANOVA was developed as follows: y = μ + construct + haplotype (construct) + e, where “y” is the relative luc activity, “μ” is the overall mean, “construct” is the construct effect (six levels: six serial 5′-deleted promoter plasmids were constructed) and “haplotype” is the haplotype effect (two levels: HG and LG) nested in the construct effect. Multiple comparison of activities between constructs was performed using Duncan’s test. Activity difference between HG and LG haplotypes was tested using *t*-test. A *p* value <0.05 was considered statistically significant. Statistical analysis was conducted using the SAS University Edition software.

## 3. Results

### 3.1. Identification of Variants in the 21 kb Region Flanking CCKAR

Over 300 variants were found within the region when all sequences were aligned to the Red Junglefowl reference genome. The HG haplotype differed from the LG haplotype by 182 variants (Figure 2; Appendix A). The alleles of the remainder were shared between HG and LG haplotypes, but were different from the reference genome. There were no non-synonymous and structural variants in the re-sequenced region except a 136 bp retrotransposable element (CR1) located 6.2 kb downstream of *CCKAR* (Appendix A). This CR1 represents a potential regulatory mutation because the elements are associated with genome stability and have the potential for acting as regulatory and coding sequences [24]. However, our previous study failed to detect an association of the CR1 with body weight (data not shown) and excluded the possibility that the CR1 element may affect *CCKAR* expression and the growth of chickens.

The screenshot of UCSC Genome Browser shows the distribution of 182 variants and the results of 76-way vertebrate conservation analysis in the 21 kb region around the *CCKAR* gene. The two alleles of each SNP are highlighted in two colors. The upper base is the allele contained in the low-growth haplotype, and the lower is present in the high-growth haplotype. Beneath the distribution plot of variants, this screenshot shows PhastCons score, PhastCons Conserved Element and Multiz Align tracts. Five variants are located in conserved noncoding regions. These variants are shown as insets seen when zooming in to the base level.

### 3.2. Defining the Core Promoter of Chicken CCKAR

Five 5′-deletion constructs showed higher promoter activities than the negative control. The pNL−328/+183 that contained the shortest fragment proximal to the transcription start site (TSS) showed the highest promoter activity among all constructs (Figure 3). With the increasing of insertion sizes, activities of promoter constructs generally decreased. However, the pNL−1646/183 showed promoter activity approaching that of pNL−799/183 (Figure 3), implying that transcriptional repression may exist between −1185 and −799 bp. pNL+303/+734 with a deleted proximal sequence around the TSS completely lost promoter activity (Figure 3).

### 3.3. Effects of Sequence Variants in the Promoter Region of CCKAR on Promoter Activity

A total of 19 SNPs were found in the −1646/+734 region, and formed into HG and LG haplotypes in the AIL (Figure 1). The HG had lower expression levels of *CCKAR* than the LG in vivo [16]. In the reporter assay, the HG showed 68% lower promoter activity than the LG on the construct pNL−1646/+183, which was in the line with expression difference in vivo (Figure 3). However, when PSNP1 and PSNP2 were deleted, the difference between HG and LG was inverted when the two haplotypes of pNL−1185/+183 were expressed (Figure 3). With deletion of PSNP3–PSNP6, the HG version had lower activity levels than the LG when the constructs pNL−799/+183 and pNL−525/+183 were expressed (Figure 3). There was no significant difference between haplotypes for the pNL−328/+183 construct (Figure 3).

### 3.4. Discovery of Conserved Sequence Variants Affecting Promoter Activity

Of 182 variants, five SNPs (CSNP1–CSNP5) were located in conserved regions with PhastCons scores of more than 0.5. Especially, CSNP1, CSNP2 and CSNP5 were embedded in the conserved elements and had PhastCons scores almost equal to 1 (Figure 2). CSNP3 alone was located in Intron 2 of *CCKAR* (Table 2; Figure 2). The others were located outside of the *CCKAR* locus (Table 2; Figure 2). The LG versions of the alleles of these SNPs were highly conserved among bird species, except CSNP4 (Table 2). Through TF prediction in silico, the five CSNP can disrupt TF motifs (Appendix A). However, promoter activity of pNLCSNP5 alone was higher than that of the negative control in the reporter assay (Figure 4). The A allele of CSNP5 was contained in the LG haplotype and was highly conserved among 52 bird species (Table 2). The activity of the A allele (LG) was 1.75-fold higher than that of the G allele (HG) (Figure 4).

## 4. Discussion

Our previous studies recognized that genetic differences in growth rate were associated with reduced satiety caused by lower expression of *CCKAR* [16]. Therefore, to identify the key *cis* elements and variants affecting *CCKAR* expression is of significance for understanding to molecular basis underlying the appetite and growth rate changes of modern chickens. This study defined the core promoter of chicken *CCKAR* and found the directions of the activity differences of three promoter and one CSNP-containing constructs were consistent with the in vivo expression patterns. These results provide an important basis for further identification of cis-regulatory variants affecting *CCKAR* expression and chicken growth.

### 4.1. Core Promoter of Chicken CCKAR Gene

A key region responsible for human *CCKAR* expression was located within 622 bp upstream from the TSS [25]. Similarly, we observed that the proximal region (−328/+183) around the TSS had the highest promoter activity in the reporter assay. Upon deleting this sequence, the pNL+303/+734 completely lost expression activity. These results suggest that the −328/+183 region is important for promoting *CCKAR* expression. In addition, we observed a significant repressive effect on expression by the −1185/+183 region. When deleting the −1185/−799 sequence, expression of pNL−799/+183 was significantly increased, implying that the −1185/−799 region should contain repressive elements. An AT-rich region between −359 and −622 bp upstream of the TSS with a repressive effect was reported in the promoter region of human *CCKAR* [25]. Similarly, an AT-rich region exists between −1378 and −776 bp upstream of the chicken *CCKAR* gene (AT contents: 71% vs. approximately 60% in adjacent regions; Appendix A). An AT-rich sequence serving as a repressive element was widely involved in the inhibition of gene expression in humans, drosophila and bacteria [26,27,28]. Coexistence of activating and repressive sequences implies that proper maintenance of *CCKAR* expression may result from the antagonistic coordination of activating and repressive effects.

### 4.2. Expression Regulation Effects of Sequence Variants in the Promoter Region of CCKAR

We detected that the HG haplotype had lower promoter activities than the LG haplotype for the pNL−1646/+183, pNL−799/+183 and pNL−525/+183 constructs—the difference being in the same direction as the results in vivo [16]. A contrary direction of haplotype difference was detected for the pNL−1185/+183 construct. This contradiction suggests that low expression of *CCKAR* in high-growth chickens may be caused by a balanced effect of expression-activating and suppressing variants. The variants located in the −1646/−1185 and −799/−328 regions contributed to low activity of the HG, and thus formed a plausible molecular basis for low expression of *CCKAR* in high-growth chickens.

A number of putative TF motifs were disrupted by these promoter variants (Appendix A). Given the roles of CCK signaling in the regulation of appetite and fat metabolism, TFs associated with energy balance may regulate *CCKAR* expression. Fork head box class O1 (FOXO1) is a well-known TF involved in appetite regulation [29]. It can stimulate appetite by activating AgRP expression and suppressing POMC transcription in hypothalamic neurons [29]. FOXO1 phosphorylation induced by the PI3K/AKT-FOXO1 pathway reverses the orexigenic effect [29]. PSNP1, PSNP3 and PSNP18 were located in the FOXO1 motif based on prediction in silico (Appendix A). Although almost nothing is known about the regulatory relationship between FOXO1 and *CCKAR*, it is possible that CCK induces satiety by the PI3K/AKT-FOXO1 signaling axis, as *CCKAR* triggers trophic effects through activating the PI3K/AKT pathways [30]. We found that low expression of *CCKAR* coincided with upregulated expression of AgRP and downregulated expression of POMC in the hypothalamus [16]. These data support an interrelationship model in which expression of *CCKAR* is downregulated due to disruption of the FOXO1 motif; absence of *CCKAR* diminishes phosphorylation of FOXO1 in hypothalamic neurons by the PI3K/AKT pathway and elicits an orexigenic effect (Appendix A).

### 4.3. Effects of Conserved Element Variants on Promoter Activity

Non-coding sequences that have critical functional roles are significantly more similar among species than would be expected under the neutral evolution model [31]. This provides a strategy for finding *cis* regulatory elements by conservation analysis of orthologous sequences. In this study we found five SNPs located in conserved sequences. These conserved sequence variants may affect *CCKAR* expression if they act as *cis* acting elements. We could not determine regulatory roles of CSNP1–CSNP4 because of the absence of activities of pNLCSNP1–pNLCSNP4 constructs in the reporter assay. However, pNLCSNP5 showed an activity difference between alleles in the expected direction in vivo [16]. Modern chicken breeds are two-fold larger in adult size than the Red Junglefowl, the ancestor of modern chickens [32]. Therefore, it is possible that functional variants affecting *CCKAR* expression and the growth of chickens derive from a recent mutation event and are exclusively present in the HG haplotype. The LG allele (A) of CSNP5 is highly conserved among 42 bird and 4 reptile species, but the HG allele (G) is rarely found in 6 bird species. Additionally, the HG allele of CSNP5 displayed a lower activity level than the LG allele in the reporter assay. Through TF prediction, we found that the HG allele can disrupt the androgen receptor (AR) motif. AR knockout was closely associated with food intake, leptin levels and adiposity in mice [33]. *CCKAR* has been developed as a promising target to treat obesity in humans [34]. The phenotypic parallels make *CCKAR* an attractive target regulated by AR. The HG allele of CSNP5 is therefore a promising candidate that may attenuate *CCKAR* expression and facilitate food intake and growth of chickens.

## 5. Conclusions

Results from serial 5′-deleted constructs suggest that the core promoter of *CCKAR* was distributed 328 bp upstream from the TSS. The parallel between results from the reporter assay and in vivo expression analysis suggests that CSNP5 and variants distributed between 328 and 1646 bp upstream form a promising molecular basis for low expression of *CCKAR* in high-growth chickens.

## Figures and Tables

**Figure 1 genes-13-01083-f001:**
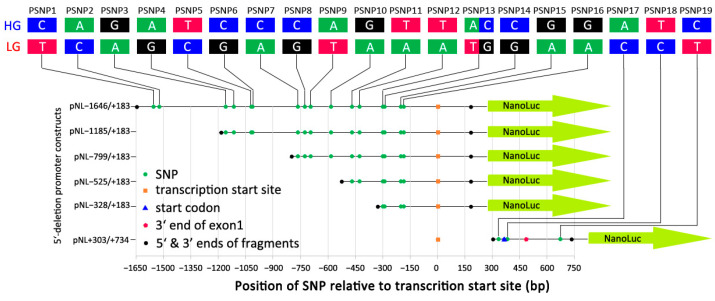
Schematic diagram of 5′-deletion constructs and distribution of variants.

**Figure 2 genes-13-01083-f002:**
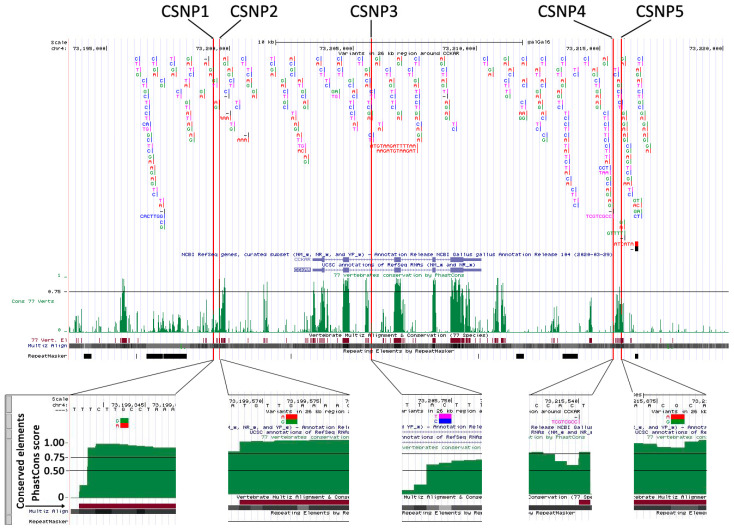
Distribution of sequence variants and results of 76-way vertebrate conservation analysis in the 21 kb region around the *CCKAR*.

**Figure 3 genes-13-01083-f003:**
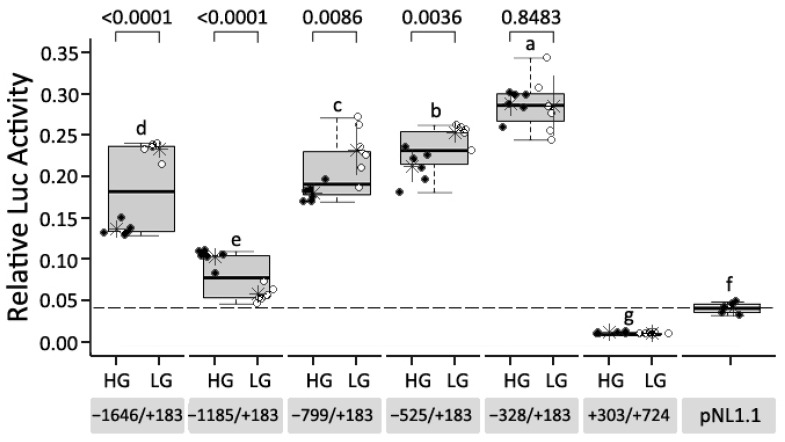
Identification of the core promoter of chicken *CCKAR* and regulatory effect analysis of sequence variants. The boxplots indicate the distribution of data from each construct. The letters on the boxplots indicate the results of multiple comparison among constructs. Different letters represent that there is significant difference (*p* < 0.05) between two constructs. Numbers at the top of each boxplot are *p*-values of significance tests for differences between haplotypes. An asterisk represents the mean of data from each haplotype, and the short lines on both sides of each asterisk represent SD. The dashed line represents background signal produced by empty pNL1.1. HG = high-growth haplotype, LG = low-growth haplotype.

**Figure 4 genes-13-01083-f004:**
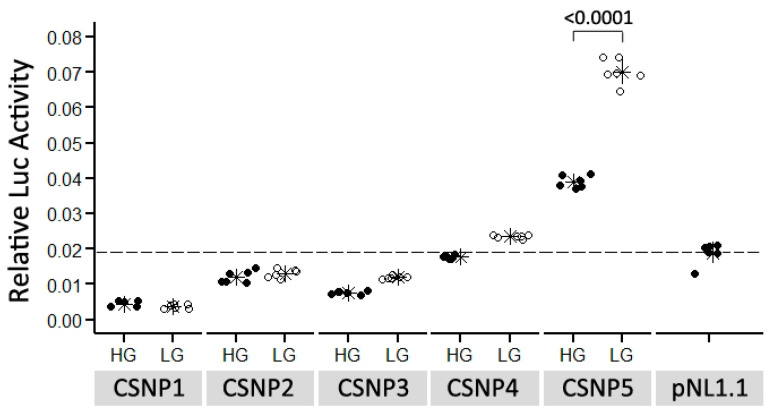
Comparison of luciferase activities of pNLCSNP between alleles. An asterisk represents the mean for each group, and the short lines on both sides of each asterisk represent SD. The number at the top of the pNLCSNP5 column is the *p*-value of the significance test. The dashed line represents the background signal produced by empty pNL1.1. HG = high-growth allele, LG = low-growth allele.

**Table 1 genes-13-01083-t001:** Annotations of promoter and conserved regions analyzed in the reporter assay, and primers for PCR amplification of target sequences.

Annotation	Positions Relative to *CCKAR* ^1^	Primer Sequence (5′-3′) ^2^	Size (bp)
Serial 5′-deleted promoter fragments	−1646/+183	F: GGGGGTACCTAGAAAGGCAGGTATGTGCT R: GTTCACTCGAGTATGCCACACGAGGCAGTTT	1829
	−1185/+1832	F: GGGGGTACCAAGCCGAATTAGGCAGCTAA R: GTTCACTCGAGTATGCCACACGAGGCAGTTT	1369
	−799/+183	F: GGGGGTACCGAAATCAGAACTTTTCAATC R: GTTCACTCGAGTATGCCACACGAGGCAGTTT	983
	−525/+183	F: GGGGGTACCCTTTTAAACAGTCAAGGCTG R: GTTCACTCGAGTATGCCACACGAGGCAGTTT	709
	−328/+183	F: GGGGGTACCGGCTTTGGATCAGGGACATG R: GTTCACTCGAGTATGCCACACGAGGCAGTTT	512
Intragenic fragment	+303/+734	F: GGGGGTACCATGGGCAGATAGTTACAAAC R: GTTCACTCGAGAAGCTGTTTACATTTTGTAC	432
CSNP-contained fragments	Upstream 4.0 kb (CSNP1)	F: GGGGGTACCCCTGCAGAGGTTCACTATGCT R:GTTCACTCGAGACCATTATTTGGGATGTTGGGA	141
	Upstream 3.8 kb (CSNP2)	F: GGGGGTACCTGATCAGCCTGAGAGAGAGTGA R: GTTCACTCGAGACTCAGCTCCCCTTTTGGAG	210
	Intron 2 (CSNP3)	F: GGGGGTACCTCACAATTTGTAAGGTTATA R: GTTCACTCGAGTCTTAAAATTCAAGAGTAAG	191
	Downstream 5.3 kb (CSNP4)	F: GGGGGTACCTCACCAACAGCCCACTACAC R: GTTCACTCGAGGGAGCTCAGACGCAACATGA	189
	Downstream 5.6 kb (CSNP5)	F: GGGGGTACCCTGCTATCTGCTGGCGTTGT R: GTTCACTCGAGGCCCTTCCAACCGCTATCTA	124

^1^ Positions are relative to the TSS or the 3′ end of *CCKAR*. ^2^ The underlined bases are *KpnI* and *XhoI* restriction sites.

**Table 2 genes-13-01083-t002:** Information of five conserved sequence variants.

Name	Description ^1^	Position Relative to *CCKAR*	LG Allele ^2^	HG Allele ^2^	Conserved Alleles among Bird Species
CSNP1	g.73199343G > A	Upstream 3.8 kb	G	A	G
CSNP2	g.73199573A > G	Upstream 4.0 kb	A	G	A
CSNP3	g.73205750T > C	Intron 2	T	C	T
CSNP4	g.73215540_73215541insTCGTCGGCC	Downstream 5.3 kb	-	TCGTCGGCC	TCGTCGGCC
CSNP5	g.73215877A > G	Downstream 5.6 kb	A	G	A

^1^ Genomic positions of SNP are given according to Galgal 6.0 chicken genome assembly. ^2^ The allele contained in the LG haplotype is defined as the LG allele, and the other in the HG haplotype is named the HG allele.

## Data Availability

Not applicable.

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
