# Peer review of "Identification of the Core Promoter and Variants Regulating Chicken CCKAR Expression"

_genes, 2022, doi:10.3390/genes13061083_

Round 1

Reviewer 1 Report

The manuscript is focused on the analysis of the upstream region of the CCKAR chicken gene, probably involved in the regulation of the food intake and, consequently of the birds’ growth. The topic is interesting and the study seems well designed. However, in my opinion, the manuscript does not adequately report such an investigation and needs improvements. Specifically, more details should be provided in the M&M and Results Sections. In particular, the latter is poor of data and sometimes not very clear. 

The omitted description of acronyms, abbreviations and genes functions furtherly reduce the readability of the manuscript.

A figure containing a map of the analyzed region with highlights of the most significant subregions would greatly aid the reader and enhance the interest of the entire study.

Similarly, the Discussion needs to be improved, by better arguing the statements in the light of the obtained results.  

Line 9: Please provide the name of the CCKAR gene in the abstract.

Line 35 and elsewhere in the text: Please always provide a brief description of the acronyms or abbreviations you use.

Lines 48-54: please specify the species to which those data are applicable.

Line 48 and elsewhere in the text: Please do not begin a sentence with an acronym.

Lines 61-63: Please rewrite that sentence to better explain the evidences supporting the presence of a cis element related to the differential expression of CCKAR.

Lines 63-64 and 65-66: Please specify if positively or negatively associated with the growth of chickens.

Lines 90-91: Please specify what regions were investigated by multiple alignments, and that could be made listed in a supplementary table. Also, please rewrite the sentience, as you probably did not align the vertebrate species but the nucleotide sequences from them.

Line 102 and elsewhere in the text: Please italicize the textual part of the restriction enzyme names.

Line 124: Please superscript the exponent.

Lines 146-147: The scientific names should be italicized.

Lines 153-155: Please specify the values associated with “construct” and “haplotype”.

Line 157 and elsewhere in the text: please italicize “P”

Lines 162-164 and 183-185: Those sentences should be included in the M&M Section rather than into Results.

Lines 169-170: Please provide more details about the CR1 element. Where is it located? Does it disrupt any coding or regulatory sequence?

Figure 1: This figure is not readable. Please upload a high-quality, high resolution image if you have not already done it.

Lines 174-181, 196-200, 204-210, and 228-231: It is not clear if those sentences are part of the figure legend or of the main text. In both cases, please provide a more accurate description of the observed results in the main text. It is not appropriate explaining the results in the figure legends only.

Lines 214-217: This sentence lacks of clarity. Please rewrite it and add some more details about the promoter activity trends within the clones with deletions.

Lines 221-225: Considering the importance the Authors confer to those SNPs, especially to CSNP5, please describe such polymorphisms.

Lines 234-236: Please rewrite this sentence as it not clear and contains several problems to be fixed.

Lines 250-252: As it is, this statement is not adequately supported by the described and discussed data. Please, provide more elements to support it.

Lines 257-259: Please better address this point.

Lines 279-287: This section mostly reviews the existing literature, but there are few references to the results of the present studies.

Lines 280-281: Please see the comment to lines 221-225.

Author Response

Dear Ms. Hazel He,

  Thanks for valuable comments you and reviewers raised about the paper. We carefully studied these comments and revised the manuscript according to these comments. Hope the revision is successful. I attach a reply to explain the changes in the manuscript.

Yours,

Zhepeng Wang

A point by point reply

Reviewer 1:

Q: Specifically, more details should be provided in the M&M and Results Sections.

A: We have supplemented necessary details in the M&M and Results sections according to your comments.

Q: Similarly, the Discussion needs to be improved, by better arguing the statements in the light of the obtained results. 

A: We have supplemented the discussion with additional evidence regarding the key points and re-written unclear sentences to improve the discussion.

Q: A figure containing a map of the analyzed region with highlights of the most significant subregions would greatly aid the reader and enhance the interest of the entire study.

A: We have provided detailed information for promoter and conserved regions in the Figure 1 and Figure 2 and reported reporter assay results in the Figure 3 and Figure 4. So we did not supplement other figures to describe the analyzed region and results.

Q: Line 9: Please provide the name of the CCKAR gene in the abstract.

A: Done

Q: Line 35 and elsewhere in the text: Please always provide a brief description of the acronyms or abbreviations you use.

A: Done

Q: Lines 48-54: please specify the species to which those data are applicable.

A: Done

Q: Line 48 and elsewhere in the text: Please do not begin a sentence with an acronym.

A: Done

Q: Lines 61-63: Please rewrite that sentence to better explain the evidences supporting the presence of a cis element related to the differential expression of CCKAR.

A: We have re-written this sentence.

Q: Lines 63-64 and 65-66: Please specify if positively or negatively associated with the growth of chickens.

A: We have specified positive or negative association of these variants with the growth of chickens.

Q: Lines 90-91: Please specify what regions were investigated by multiple alignments, and that could be made listed in a supplementary table. Also, please rewrite the sentence, as you probably did not align the vertebrate species but the nucleotide sequences from them.

A: We have re-written the sentence to clarify the problem.

Q: Line 102 and elsewhere in the text: Please italicize the textual part of the restriction enzyme names.

A: Done

Q: Line 124: Please superscript the exponent.

A: Done

Q: Lines 146-147: The scientific names should be italicized.

A: Done

Q: Lines 153-155: Please specify the values associated with “construct” and “haplotype”.

A: “construct” and “haplotype” represent two treatment effects in the ANOVA model. “haplotype” is nested in the “construct” effect. The number of values for each variable is indicated in the text, 6 for the constructs and 2 for the haplotype.

Q: Line 157 and elsewhere in the text: please italicize “P”

A: Done

Q: Lines 162-164 and 183-185: Those sentences should be included in the M&M Section rather than into Results.

A: We have moved these contents to the M&M section.

Q: Lines 169-170: Please provide more details about the CR1 element. Where is it located? Does it disrupt any coding or regulatory sequence?

A: We have provided position information of CR1 element in the main text. The physical position of CR1 in the chromosome 4 is listed in the Table S2. We have provided an explanation of why the mutation was not included in the reporter assay.

Q: Figure 1: This figure is not readable. Please upload a high-quality, high resolution image if you have not already done it.

A: Done

Q: Lines 174-181, 196-200, 204-210, and 228-231: It is not clear if those sentences are part of the figure legend or of the main text. In both cases, please provide a more accurate description of the observed results in the main text. It is not appropriate explaining the results in the figure legends only.

A: These sentence are figure legends. There are corresponding explanations in the Results section.

Q: Lines 214-217: This sentence lacks of clarity. Please rewrite it and add some more details about the promoter activity trends within the clones with deletions.

A: We have re-written the sentence to clarify the confused description.

Q: Lines 221-225: Considering the importance the Authors confer to those SNPs, especially to CSNP5, please describe such polymorphisms.

A: We have supplemented Table 2 to describe position, allele and conserved alleles of five CSNP. We have provided a detailed description of the association of CSNP5 alleles with the growth of chickens and conservation analysis results for the polymorphism in bird species.

Q: Lines 234-236: Please rewrite this sentence as it not clear and contains several problems to be fixed.

A: We have re-written the sentence.

Q: Lines 250-252: As it is, this statement is not adequately supported by the described and discussed data. Please, provide more elements to support it.

A: We supplemented additional evidence from studies on regulatory roles of AT-rich elements to support the view of balancing regulatory effect. We have also re-written those sentences to increase the quality of the discussion.

Q: Lines 257-259: Please better address this point.

A: In the reported assay we detected a contrary difference between HG and LG for pNL-1185/183. In the case that we lack experimental evidence about regulatory effects of these promoter variants, it is difficult to better address this point besides the hypothesis that low expression of CCKAR in high-growth chickens may results from balancing effects of multiple promoter variants.

Q: Lines 279-287: This section mostly reviews the existing literature, but there are few references to the results of the present studies.

A: We have re-written the contents to discuss the results.

Q: Lines 280-281: Please see the comment to lines 221-225.

A: Done

Reviewer 2 Report

The manuscript entitled " Identification of core promoter and variants regulating chicken CCKAR expression" represents a considerable amount of work. The following comments need to be addressed.

-          Lines 52 and 54: Please change CCKAR to CCKAR.

Author Response

Q: Lines 52 and 54: Please change CCKAR to CCKAR

A: CCKAR in these sentence represents the receptor protein rather than gene. So CCKAR is appropriate.

Reviewer 3 Report

Authors amplifeid by PCR sequenced the 21 kb region around the chicken CCKAR gene. Authors identified sequence variants that may be asociated with high or low growth phenotype. Nineteen variants are localized in CCKAR promoter region and may be associated with differences in chicken growth rate. Moreover Authors defined experimentally the CCKAR core promoter region and proposed a model expaining the role FOXO1 trans-factor in appetite regulation. Research is well organized and perfomed. Obtained data suport conclusion.

Some minor comments should be addressed.

POMC/CART and CCK abbreviations should be explained

Section 2.1

Name of DNA polymerase used to amplify the 21 kb DNA fragment. Conditions of PCR reaction and name/manufacturer of PCR equipment should be provided.

Section 3.4

Authors wrote of 5 CSNP (1-5). Only two of them CSNP1 and CSNP2 are localized in CCKAR promoter region (Table S1). However, Figure S1 contains information of trans-factors binding to CSNP3-5, that are beyond the promoter region. Authors should explain it.

Author Response

Q: POMC/CART and CCK abbreviations should be explained

A: Done

Q: Name of DNA polymerase used to amplify the 21 kb DNA fragment. Conditions of PCR reaction and name/manufacturer of PCR equipment should be provided.

A: We have supplemented relevant contents according to your suggestion.

Q: Authors wrote of 5 CSNP (1-5). Only two of them CSNP1 and CSNP2 are localized in CCKAR promoter region (Table S1). However, Figure S1 contains information of trans-factors binding to CSNP3-5, that are beyond the promoter region. Authors should explain it.

A: CSNP3, CSNP4 and CSNP5 are located in the Intron 2, 5.3 kb and 5.6 kb downstream of CCKAR gene. They can act as cis acting elements beyond a classical promoter region to regulate CCKAR expression, i.e. enhancers. So TF prediction for the variants at the outside of promoter was included in the study.